# E-Cigarette or Vaping Product Use-Associated Lung Injury: A Comprehensive Review

**DOI:** 10.3390/ijerph22050792

**Published:** 2025-05-17

**Authors:** Mohammad Asim Amjad, Daniel Ocazionez Trujillo, Rosa M. Estrada-Y-Martin, Sujith V. Cherian

**Affiliations:** 1Divisions of Critical Care, Pulmonary and Sleep Medicine, Department of Internal Medicine, University of Texas Health-McGovern Medical School, 6431 Fannin Street, MSB 1.434, Houston, TX 77030, USA; mohammad.asim.amjad@uth.tmc.edu (M.A.A.); rosa.m.estrada.y.martin@uth.tmc.edu (R.M.E.-Y.-M.); 2Department of Diagnostic and Interventional Imaging, University of Texas Health-McGovern Medical School, Houston, TX 77030, USA; daniel.ocazioneztrujillo@uth.tmc.edu

**Keywords:** e-cigarettes, electronic cigarette or vaping product-associated lung injury, hypoxemic respiratory failure, ARDS

## Abstract

E-cigarette or vaping product use-associated lung injury (EVALI) is a critical and potentially fatal form of lung injury that gained considerable public health concern in 2019. The use of e-cigarettes and vaping products is causally associated with EVALI, a condition characterized by a constellation of respiratory symptoms, such as coughing, shortness of breath, and chest pain. This comprehensive narrative literature review explores the complexities of EVALI, including its association with the structure and composition of e-cigarettes and its epidemiology, pathogenesis, clinical and radiological manifestations, management strategies, and public health implications. Moreover, it uncovers the long-term repercussions of EVALI and underscores the ongoing research endeavors designed to mitigate and comprehend the risks associated with using e-cigarettes.

## 1. Introduction

The prevalence of e-cigarette and vaping product use has skyrocketed in recent years, particularly among adolescents and young adults. E-cigarette or vaping product use-associated lung injury (EVALI) is a serious condition characterized by respiratory symptoms and lung damage associated with the use of e-cigarettes and vaping products. Dai and Leventhal (2019) reported that the prevalence of e-cigarette use among adults in the United States increased from 3.7% in 2014 to 4.9% in 2018 [1]. The increase has been even more dramatic among youth, with Cullen et al. (2019) finding that 27.5% of high school students and 10.5% of middle school students reported current e-cigarette use in 2019 [2]. The significant impact on youth has continued to be a major concern, with studies like Tirmizi et al. (2024) characterizing EVALI as an evolving pediatric public health crisis and highlighting ongoing issues with surveillance and prevention in this vulnerable population [3]. Initially marketed as a less harmful alternative to traditional cigarettes, these products have been increasingly associated with a myriad of risks, including various pulmonary complications, among which EVALI is a prominent and severe example [4,5,6]. The initial reports of EVALI emphasized the severity of EVALI and its varied manifestations, including GI symptoms like nausea, vomiting and abdominal pain. Moreover, hypoxemic respiratory failure was seen in a considerable proportion of patients, with mechanical ventilation needed up to one-third of such patients. It is important to note that Vitamin E acetate was identified as a primary toxicant in EVALI cases, predominantly found in marijuana-vaping devices.

EVALI, a severe form of vaping-associated lung injury, necessitates prompt recognition and appropriate medical intervention to mitigate potential long-term pulmonary complications and mortality. The specific objectives of this comprehensive literature review are:Provide a comprehensive overview of the epidemiology and public health impact of EVALI.Detail the current understanding of the pathogenesis of EVALI, including the role of various e-liquid components and device characteristics.Describe the clinical, radiological, and pathological features characteristic of EVALI.Summarize current diagnostic approaches and clinical management strategies.Discuss the known long-term consequences and ongoing research directions related to EVALI.

The information presented in this literature review was gathered through searches of prominent biomedical databases, including PubMed, Google Scholar, and Scopus, using keywords such as ‘EVALI’, ‘e-cigarette associated lung injury’, ‘vaping lung injury’, ‘vitamin E acetate’, and ‘THC vaping’ focusing on peer-reviewed articles, consensus statements, and reports from public health organizations published up to early 2025. The selection aimed to be comprehensive, covering seminal works, recent advancements, and diverse perspectives on the topic, with articles chosen based on their relevance to the predefined sections of this review. No formal quality assessment or specific inclusion/exclusion criteria typical of systematic reviews were applied, as the aim was to provide a broad and comprehensive overview.

## 2. Epidemiology

The 2019–2020 outbreak of EVALI predominantly affected the United States, although cases were reported globally [4,5]. By 15 October 2019, 1479, EVALI cases were confirmed across 49 states, the District of Columbia, and one US territory. The outbreak disproportionately impacted young people, with a median age of 24 years, highlighting the heightened use within this age group [4]. The Centers for Disease Control and Prevention (CDC) reported a substantial increase in emergency department visits related to e-cigarette and vaping product use in August 2019, reaching a peak in September 2019 before gradually declining. By February 2020, the CDC had identified 2807 hospitalized EVALI patients with nonfatal cases and 60 fatal cases [7]. The CDC played a crucial role in detecting and monitoring EVALI cases by establishing surveillance systems to track trends and disseminate timely information to healthcare providers and the public [4]. These efforts, combined with public health interventions, contributed to a reduction in EVALI incidence. Chatham-Stephens et al. (2019) observed that the weekly incidence of EVALI peaked in mid-September 2019, followed by a rapid decline with increased public awareness and regulatory scrutiny, particularly regarding vitamin E acetate in illicit THC-containing products [8]. Despite this significant decline, sporadic cases continue to be reported, albeit at a much lower frequency. Recent reviews, such as that by Alqahtani et al. (2025), reiterate that despite the overall decline after the 2019–2020 outbreak and removal of vitamin E acetate from many products, EVALI cases are still being detected, emphasizing the need for ongoing vigilance [9].

The CDC’s initial surveillance efforts were crucial in tracking the outbreak and informing public health responses. However, as highlighted by Rebuli et al. (2023), “continued surveillance remains critical for identifying new cases, monitoring trends, and informing public health strategies” [5]. This underscores the ongoing need for vigilance and comprehensive public health efforts to prevent future cases, especially as vaping products and user behaviors evolve. Understanding the evolving landscape of e-cigarette use, including new devices and substances, remains vital for effective surveillance and prevention initiatives.

## 3. Structure of E-Cigarettes

E-cigarettes generally comprise three primary components: a battery, a heating element, and a cartridge or tank that holds e-liquid (Figure 1). The battery energizes the heating element, which vaporizes the e-liquid to generate an aerosol for inhalation by the user [10]. E-liquids generally comprise a foundation of propylene glycol and/or vegetable glycerin, nicotine, flavorings, and assorted additives. Margham et al. (2021) examined the chemical intricacy of e-cigarette aerosols and discovered that they encompass a diverse array of volatile organic chemicals, including aldehydes, ketones, and hydrocarbons [10].

Although propylene glycol and vegetable glycerin are typically regarded as safe for consumption, their prolonged impact on the respiratory system when inhaled is still uncertain [11,12]. Nicotine, a potent stimulant with a strong addictive potential, can impact cardiovascular health and may lead to nicotine dependency. Beyond its addictive properties, nicotine has significant effects on the cardiovascular system. It can increase heart rate, blood pressure, and cause vasoconstriction, which can contribute to the development and progression of cardiovascular diseases. E-cigarette use is also associated with increased oxidative stress and inflammation, both of which play a crucial role in the pathogenesis of cardiovascular disease [13,14,15]. The formulation of e-liquids is significantly diverse, and the long-term health implications of numerous components remain predominantly uncertain. Moreover, the heating process may produce novel chemical compounds, complicating the evaluation of potential health risks [16,17].

The variety of e-cigarette devices, such as vape pens, mods, and pod systems (different types of e-cigarette devices, all of which share the basic structure illustrated in Figure 1), complicates the assessment of their possible health impacts. Diverse devices can provide disparate quantities of nicotine and other substances, and the temperature at which the e-liquid is vaporized can also affect the aerosol’s composition [17]. Furthermore, the proliferation of counterfeit and unregulated products creates apprehension regarding the quality and safety of e-cigarettes accessible to users.

## 4. E-Liquid Components

E-liquids contain a complex mixture of substances, and research has shown that different components can contribute to various forms of lung injury (Table 1). While some components have been specifically linked to EVALI, others are associated with a broader range of respiratory illnesses. The US Food and Drug Administration (FDA) classified many flavorings as “generally recognized as safe” (GRAS) [16]. However, this designation only applies to foods that are eaten, not to products that are inhaled. The respiratory tract was found to be contaminated by at least 65 individual flavoring ingredients in flavored e-liquids, which caused toxicity by inducing cytotoxicity, generating reactive oxygen species, and impairing clearance mechanisms (Table 2) [17]. Cinnamaldehyde, vanillin, menthol, and other flavoring chemicals have been found to cause the most toxicity in in vitro investigations [17,18]. The extensive range of items on the market and the ongoing changes to e-cigarette and vaping devices make it difficult to assess the biological risk of vaping and specific e-liquids fully [6].

Around the turn of the millennium, concerns arose regarding the safety of diacetyl, a flavoring agent used in e-liquids to impart a buttery taste [33]. This chemical was linked to the development of bronchiolitis obliterans, a serious lung condition also known as “popcorn lung”, in workers at a microwave popcorn factory. This condition involves inflammation and scarring within the lung, leading to obstructed airways [21]. Furthermore, research indicates that propylene glycol, another common e-liquid ingredient that helps mix other components, may also contribute to airway damage. Studies suggest it can harm the delicate lining of the airways and hinder cell repair [11]. This damage may be particularly pronounced in e-cigarette users who already have chronic obstructive pulmonary disease (COPD). Lastly, vegetable glycerin, which carries the active compounds in e-liquids, has been found to disrupt normal function in the nasal passages. This disruption can lead to thicker mucus, potentially increasing the risk of inflammation and impaired airway function [17].

It is important to consider that vaping products encompass a range of substances, most notably nicotine and cannabis. Nicotine-based e-liquids typically utilize propylene glycol and vegetable glycerin as base components, along with nicotine and flavorings. Cannabis vaping products, on the other hand, often contain tetrahydrocannabinol (THC) or other cannabinoids as their active ingredients. A key difference lies in the common use of vitamin E acetate as a thickening agent in cannabis-containing e-liquids, which has been strongly linked to EVALI [34,35].

Vitamin E acetate, often used as a thickening agent in cannabis-containing e-liquids, has emerged as a key factor in the EVALI outbreak. The CDC found vitamin E acetate present in the bronchoalveolar lavage of most EVALI patients, strongly suggesting a link between this substance and the onset of the illness [35,36]. Studies show that it can impair breathing and, when heated, it decomposes into harmful compounds like ketene, alkene, and benzene, all of which can damage lung tissue [37]. Furthermore, research indicates that vitamin E acetate interferes with the normal function of lung surfactant, a substance crucial for maintaining proper lung function [38]. This disruption can increase surface tension within the lungs, potentially triggering inflammation and contributing to the development of EVALI.

The 2019–2020 EVALI outbreak was linked to the vaping of THC-containing products, particularly those adulterated with vitamin E acetate [28,29,39]. Vitamin E acetate, while safe for ingestion or topical application, was widely used as a thickening agent in illicit THC vaping liquids, likely to dilute THC oil without significantly altering its viscosity [34,35]. The CDC’s findings consistently identified vitamin E acetate in bronchoalveolar lavage fluid from EVALI patients and in product samples associated with the outbreak, pointing to it as the primary chemical of concern [28,36]. These dangerous formulations were distributed through illegal market or informal sources rather than approved, regulated marketplaces [40]. Such illicitly sourced vape products are inherently risky due to their unregulated nature, leading to significant variability in product composition, contaminant levels, and the concentrations of harmful additives like vitamin E acetate [5,41]. This lack of oversight means that black market vape products can differ substantially not only in their primary constituents but also in the presence of other unknown toxicants, potentially varying from batch to batch or across different informal supply chains, thereby posing unpredictable and severe risks to users [5]. The persistence of such unregulated products in the market underscores the importance of continued vigilance, as they are likely to contribute to the sporadic EVALI cases that are still being reported even after the peak of the initial outbreak [3,9].

The precise mechanisms by which different e-liquid components cause lung injury are still being investigated. EVALI appears to have a multifactorial etiology, with vitamin E acetate playing a significant role [35]. However, other components may contribute to the severity or specific characteristics of the injury. Further research is needed to fully understand the long-term effects of various e-liquid constituents and their potential to cause a range of pulmonary diseases. Additionally, THC, the psychoactive component in cannabis, is often added to e-liquids. When heated, it can degrade into toxic substances like methacrolein and benzene [37].

**Table 2 ijerph-22-00792-t002:** Factors contributing to EVALI.

Pathogenic Factor		Function	Mechanism of Action	References
Vitamin E acetate (alpha-tocopherol acetate)		Thickening agent, particulary in cannabis-containing e-liquids	Disrupts surfactant function	[28,42]
Flavoring Agents	Cinnamaldehyde	Provide flavor to e-liquids	Direct cellular toxicity Airway irritation Suppression of macrophage phagocytosis Decreased neutrophil oxidative burst	[43,44]
	Vanillin			
	Ethyl vanillin			
	Menthol			
Heavy metals		Various (present as contaminants)	Increases oxidative stress	[10]
Propylene glycol		Base of e-liquids, solvent	Direct cytotoxic Generates carcinogenic compounds	[45,46]
Diacetyl		Flavoring agent (buttery flavor)	Airway inflammation Associated with bronchiolitis	[33]
Vegetable glycerin (glycerol/glycerin)		Base of e-liquids	Increase mucin expression	[12,47]

## 5. Pathogenesis of Vaping-Induced Lung Injury

E-cigarette use, or vaping, has been linked to a spectrum of pulmonary complications, including EVALI and other lung injuries (Table 1) [19,20,30]. While the exact mechanisms underlying these conditions remain an active area of investigation, a growing body of evidence points to a complex interplay of chemical and inflammatory processes [30]. To better understand how vaping leads to lung injury, it is helpful to consider several potential pathways. Table 3 provides an overview of these potential mechanisms.

Crotty Alexander et al. (2020) proposed two hypotheses regarding the mechanisms of injury [59]. First, they suggested that a chemical inhaled from the aerosol generated by vaping or dabbing is directly cytotoxic to specific lung cells. This cytotoxicity results in cellular necrosis, neutrophilic inflammation, and collateral damage. Second, they proposed a ’two-hit’ phenomenon. In this scenario, inhalation of the base components of e-liquids, such as propylene glycol and glycerin, from e-devices induces alterations in the homeostatic state of lung immune cells. This disruption of equilibrium leads to significant inflammation upon the introduction of a typically well-tolerated inhalant into the lungs [59]. The following sections will delve into these and other key mechanisms contributing to vaping-related lung injury.

### 5.1. Direct Cellular Injury and Inflammation

Upon inhalation, aerosolized e-liquids deposit throughout the respiratory system, exposing the delicate lung epithelium to potentially harmful chemicals [48]. Components like menthol, ethyl maltol, and volatile organic compounds generated at high temperatures can trigger inflammation, a key driver of vaping-related lung injury [37,49,50]. This inflammatory response is further amplified by reactive oxygen species (ROS) generated by e-cigarette aerosols, leading to cellular apoptosis through ROS-mediated autophagy [60,61]. This mechanism has also been implicated in emphysema, a chronic obstructive pulmonary disease characterized by alveolar destruction [16].

### 5.2. Bronchiolitis Obliterans

Chronic airway centric inflammation and fibrosis can culminate in bronchiolitis obliterans, a severe and often irreversible condition marked by bronchiolar smooth muscle hypertrophy, peribronchiolar inflammation, intraluminal mucus accumulation, and ultimately, fibrotic scarring [51]. Bronchiolitis obliterans carries a poor prognosis, with many patients experiencing a progressive decline in pulmonary function, potentially requiring mechanical ventilation or lung transplantation [62].

### 5.3. Heavy Metal Toxicity

E-cigarette aerosols, particularly those generated from pod-type devices, contain heavy metals such as chromium, nickel, and lead [16]. These metals originate from the degradation of e-cigarette device components, like filaments and coils, when exposed to acidic e-liquids. While the pulmonary consequences of heavy metal exposure in e-cigarette users are not fully elucidated, these metals are known risk factors for various respiratory diseases, including bronchitis, asthma, COPD, and lung cancer [54].

### 5.4. Lipid-Laden Alveolar Macrophages

A hallmark of EVALI is the presence of lipid-laden alveolar macrophages, also known as foam cells, in bronchoalveolar lavage fluid [22,45]. While these macrophages are crucial in clearing inhaled particles and pathogens, their accumulation in EVALI suggests a dysregulation of lipid metabolism and immune homeostasis [23,63]. Although lipid-laden macrophages are not specific to EVALI and can be observed in other pulmonary conditions, their presence in EVALI patients underscores the role of alveolar macrophages in the pathogenesis of vaping-related lung injury [24].

### 5.5. Disruption of the Alveolar–Capillary Barrier

Acute lung injury in EVALI is characterized by damage to both alveolar epithelial cells and pulmonary vascular endothelial cells, compromising the integrity of the alveolar–capillary barrier [25]. This disruption leads to fluid accumulation in the alveoli (pulmonary edema), neutrophil infiltration, and the release of cytotoxic and pro-inflammatory mediators, culminating in respiratory distress [49].

### 5.6. Vitamin E Acetate and Surfactant Dysfunction

Vitamin E acetate, a common additive in e-liquids, particularly those containing THC, has emerged as a key contributor to EVALI [26]. Vitamin E acetate is thought to interfere with pulmonary surfactant function, leading to alveolar collapse and inflammation [6,58]. Studies have detected Vitamin E acetate in the bronchoalveolar lavage fluid of a majority of EVALI patients, further supporting its role in the pathogenesis of this condition [26,27,28].

### 5.7. Immune Dysregulation

E-cigarette components can impair respiratory immune cell function, rendering the lungs more susceptible to injury and infection. E-cigarette aerosols induce oxidative stress and inflammation, contributing to tissue damage and impaired lung function [43,44].

The pathogenesis of vaping-related lung injury is thus a complex process involving multiple interacting pathways. Further research is needed to fully elucidate the precise mechanisms involved and develop effective prevention and treatment strategies.

## 6. Clinical Features

The onset of EVALI symptoms can be variable, ranging from days to weeks after vaping. EVALI generally manifests with a range of respiratory symptoms, such as cough, dyspnea, thoracic discomfort, and pyrexia [29,64,65,66]. Numerous case series have offered comprehensive accounts of the clinical characteristics of EVALI. Layden et al. (2020) documented 53 EVALI patients in Illinois and Wisconsin, identifying cough (97%), shortness of breath (85%), and chest pain (56%) as the predominant symptoms [29]. Fatigue, fever, and symptoms related to the gastrointestinal tract were mentioned among the other prevalent symptoms. Similar respiratory symptoms were reported in 80 EVALI patients in Utah by Blagev et al. (2019), with 31% needing mechanical ventilation [67]. Maddock et al. (2019) reported on 19 EVALI patients in Wisconsin, emphasizing the presence of lipid-laden macrophages in bronchoalveolar lavage (BAL) fluid [24]. However, the pathological findings in lung biopsies from 17 EVALI patients were described by Butt et al. (2019), who observed diffuse alveolar injury and the presence of lipid-laden macrophages [30]. The severity of EVALI can range from mild to life-threatening. Some patients may require hospitalization and intensive care, including mechanical ventilation [29,52].

## 7. Radiology

Radiological imaging, particularly computed tomography (CT), has proven essential in characterizing the pulmonary manifestations of EVALI [64,68,69]. Ground-glass opacities (GGOs) are the most frequently reported finding on CT scans, often accompanied by consolidations, subpleural sparing, and septal thickening (Figure 2, Figure 3 and Figure 4). These findings suggest a diffuse alveolar injury pattern with varying degrees of inflammation and airspace filling (Figure 5). While GGOs and consolidations reflect acute inflammatory processes, subpleural sparing and septal thickening may indicate the development of fibrosis and chronic lung changes.

Kligerman et al. (2020) reported a series of EVALI cases with diverse CT findings, highlighting the heterogeneity of imaging manifestations. Their case series illustrated the spectrum of EVALI severity, ranging from complete resolution with treatment to rapid progression and fatal outcomes [64]. This heterogeneity is further reflected in observations from other studies. Layden et al. (2020) emphasized bilateral alveolar opacities as a standard feature, while Pajak et al. (2020) noted subpleural sparing alongside opacities and consolidation [29,70].

Other studies have confirmed the predominance of GGOs, consolidations, and subpleural sparing in EVALI patients. Notably, Kalininskiy et al. (2020) and Aberegg et al. (2019) observed the resolution of CT abnormalities in patients treated with antibiotics and/or corticosteroids, suggesting the potential for lung recovery in some cases [71,72]. Furthermore, Panse et al. (2021) contributed valuable longitudinal data, showing how EVALI’s CT appearance evolves, which is crucial for understanding disease progression and potential chronic changes [31].

Although CT is the primary imaging modality for EVALI assessment, other techniques like positron emission tomography (PET) and pulmonary functional MRI may provide complementary information about lung function and metabolic activity [68,73]. PET with specific tracers can assess ventilation, perfusion, inflammation, and the deposition of inhaled substances. Pulmonary functional MRI, including hyperpolarized gas MRI and oxygen-enhanced MRI, offers noninvasive methods to evaluate ventilation and perfusion and is sensitive to early disease changes. Kizhakke Puliyakote et al. (2023) used MRI to assess ventilation-perfusion mismatch in asymptomatic e-cigarette users. They found impaired matching due to alterations in both ventilation and perfusion, with a degree of disruption comparable to that observed in patients with chronic obstructive pulmonary disease (COPD) [74].

Beyond the immediate effects of EVALI, several studies have explored the long-term pulmonary consequences and functional changes associated with vaping. Eddy et al. (2021) reported a case of a teenage boy who developed persistent, chronic airflow limitations and gas trapping requiring mechanical ventilation and extracorporeal membrane oxygenation (ECMO) following EVALI, highlighting the potential for severe and irreversible lung damage in young individuals [75]. Nyilas et al. (2023) examined the immediate effects of electronic nicotine delivery systems (ENDS) exposure and tobacco smoke on lung ventilation and perfusion using functional MRI and lung function tests [73]. They observed increased local perfusion in participants who used ENDS after exposure but no changes in lung function compared to baseline. Wetherill et al. (2024) used PET to quantify inducible nitric oxide synthase expression to characterize oxidative stress and lung inflammation in vivo. ENDS users showed greater expression than cigarette smokers and controls [76,77]. Prospective longitudinal studies with long-term follow-up are needed to determine the clinical significance of residual CT abnormalities and to assess the potential for permanent lung damage in EVALI patients. Such studies will contribute to a better understanding of the natural history of EVALI and inform strategies for optimizing patient care.

## 8. Pathology

The histopathology of EVALI is nonspecific and often presents challenges in diagnosis. Lung biopsies typically demonstrate a pattern of organizing pneumonia (OP) [30,31], mainly when obtained during the subacute phase of illness. Less frequently, patterns of acute fibrinous pneumonitis or diffuse alveolar damage are observed, usually in patients with more acute or severe presentations. While no specific histopathologic features definitively diagnose EVALI, several clues can suggest potential causes. Foamy macrophages are commonly present (Figure 5), indicating toxic injury, and changes are often centered around the bronchioles [32]. Most cases exhibit granular proteinaceous or pigmented debris within the injured areas. Neutrophilic inflammation can be prominent, mimicking infection or acute interstitial pneumonia, highlighting the need for careful evaluation. Pathologists should prioritize the assessment for infection using special stains for microorganisms and cultures [32].

Early reports of EVALI described lipid-laden macrophages in BAL fluid, identifiable by oil red O (Figure 6) staining [28]. While initially promising as a potential diagnostic marker, this assay lacks specificity for EVALI. Oil red O-positive macrophages can be found in various conditions, including COVID-19, other infections, drug reactions, and autoimmune disorders, all of which involve the accumulation of lipid-containing macrophages from cell membrane breakdown [78]. Due to its lack of specificity, technical challenges, and limited data on sensitivity for EVALI, oil red O staining should not be used as a sole diagnostic tool [52,78].

## 9. Diagnosis

EVALI presents a diagnostic challenge due to the absence of a single, definitive test. Identifying the condition relies on a combination of clinical signs, symptoms, and a history of vaping. In 2019, the CDC established criteria to help standardize reporting during the EVALI outbreak (see Table 4, adopted from) [5,79]. These criteria emphasize recent e-cigarette use (within 90 days) coupled with pulmonary infiltrates or opacities visible on chest radiography (X-ray) or computed tomography (CT) scan. Crucially, the CDC criteria also require ruling out alternative diagnoses, particularly infections. While the CDC criteria were invaluable for surveillance during the outbreak, they were not intended as strict diagnostic guidelines for individual patients. The diagnostic complexities are further highlighted by Alqahtani et al. (2025), who discuss the challenges of differentiating EVALI from other respiratory illnesses, including COVID-19, particularly when relying on non-specific symptoms and imaging findings [9]. As emphasized in the 2023 ATS statement by Rebuli, ongoing research and clinical experience are essential for refining diagnostic approaches and management strategies for EVALI [5]. Diagnosing EVALI involves a comprehensive medical assessment to exclude other potential causes of the patient’s respiratory symptoms [5]. Typically, it includes a detailed patient history, focusing on vaping habits, symptom onset, and any other medical conditions. A thorough physical examination helps assess overall health and identify specific signs of respiratory distress. Imaging studies, such as chest X-rays or CT scans, are essential to visualize lung abnormalities, including ground-glass opacities often seen in EVALI. While not specific to EVALI, laboratory tests can reveal clues like elevated white blood cell count, inflammation markers (like C-reactive protein), and possibly eosinophilia. These findings must be interpreted cautiously, as they can occur in other conditions [53].

Ruling out infections is critical for EVALI diagnosis and may involve tests for respiratory viruses, bacterial cultures, and PCR testing [55]. BAL fluid analysis in EVALI often shows increased neutrophils and lipid-laden macrophages. Still, these findings are not unique to EVALI and need careful interpretation within the patient’s overall clinical picture [24,45]. It is important to note that the CDC’s definition does not consider the specific type of e-liquid used, nor does it address chronic respiratory issues potentially caused or worsened by vaping.

## 10. Clinical Management

The clinical management of EVALI primarily focuses on supportive care and symptom relief, as no randomized clinical trials have yet evaluated specific therapies [6]. Most patients hospitalized with EVALI survive, with a reported mortality rate of 2.4% based on CDC data [40]. However, long-term outcomes remain poorly understood, making defining the optimal management approach challenging. Emerging studies suggest that a significant proportion of EVALI survivors experience persistent respiratory issues, cognitive impairment, and mood disturbances despite often discontinuing the use of e-cigarettes [7,56,64].

Respiratory support is a cornerstone of EVALI management. This can range from supplemental oxygen, titrated to maintain SpO_2_ ≥ 90%, to noninvasive ventilation (NIV) for patients with moderate respiratory distress who do not respond adequately to oxygen alone, with careful monitoring for worsening respiratory status. Invasive mechanical ventilation is indicated for patients with severe respiratory failure, including those with refractory hypoxemia or hypercapnia, employing lung-protective ventilation strategies to minimize barotrauma. Extracorporeal membrane oxygenation (ECMO) is reserved for select cases of severe, refractory respiratory failure, typically in specialized centers with expertise in ECMO management [55]. Antibiotic therapy is often initiated, especially in the early stages of hospitalization, as EVALI can mimic bacterial or viral pneumonia [72]. Broad-spectrum antibiotics are typically used until infectious etiologies are ruled out. Corticosteroids are often used, with some case series reporting their use in 67–90% of patients [29,71]. However, the dosing and duration of corticosteroid therapy vary widely, and many patients recover with supportive care alone [72,80]. When using corticosteroids, clinicians should consider the potential for immunosuppression and the increased risk of secondary infections; the need for close monitoring for hyperglycemia, especially in patients with diabetes; the potential for psychiatric side effects, particularly with high doses; and a slow taper of corticosteroids is recommended to prevent rebound symptoms.

A crucial aspect of EVALI management is the complete cessation of e-cigarette use. Continued vaping has been associated with recurrent EVALI and respiratory failure [67]. However, achieving cessation can be challenging due to the addictive nature of nicotine and THC, particularly in adolescents and young adults, where data on effective cessation strategies are limited [81]. The CDC recommends offering or referring patients to cessation services in both inpatient and outpatient settings.

Given the complexities of EVALI management, healthcare providers should adhere to the following guidelines. Promptly assess patients with respiratory symptoms who report e-cigarette use, including a detailed history, physical examination, and appropriate laboratory and radiological investigations. This includes obtaining a thorough history of e-cigarette product usage, including type of device, e-liquid components, and duration of use. Implement supportive care, including respiratory support as needed, and consider corticosteroids based on clinical judgment, weighing the risks and benefits. Closely monitor patients for respiratory distress, secondary infections, pneumothorax, and other potential complications. Serial monitoring of oxygen saturation, respiratory rate, and chest radiographs is essential. Provide or refer patients to smoking cessation services, emphasizing the importance of complete cessation of e-cigarette use. Utilize evidence-based cessation strategies, including nicotine replacement therapy and behavioral counseling. Report EVALI cases to relevant public health authorities to contribute to ongoing surveillance and research.

Despite lacking clinical trials and long-term outcome data, the CDC published a management algorithm for EVALI in 2019. Many medical centers developed similar regional treatment algorithms based on their clinical experience [71]. However, with the decline in EVALI cases and the emergence of the COVID-19 pandemic, interventional trials for EVALI have not been conducted and are not currently planned.

## 11. Public Health Interventions

EVALI carries significant public health implications, necessitating comprehensive strategies to mitigate the risks associated with e-cigarettes and vaping products [41,57]. Public education campaigns are crucial to raise awareness about the potential dangers of these products, particularly among youth and young adults, while dispelling misconceptions about their safety. Product regulations should focus on restricting flavored e-cigarettes, limiting nicotine concentrations and other harmful chemicals in e-liquids, and ensuring product safety and quality [5]. Enforcing age restrictions is vital to prevent youth access to e-cigarettes [41]. Continued surveillance is essential to monitor trends, identify new EVALI cases, and inform public health interventions [81]. Recent analyses, such as Tirmizi et al. (2024), emphasize the continued need for targeted public health interventions and robust surveillance systems specifically for EVALI cases [3].

## 12. Long-Term Consequences of EVALI

While the acute effects of EVALI are well-documented, long-term consequences remain an active area of investigation [81]. Some patients experience persistent respiratory symptoms, such as cough, shortness of breath, and reduced lung function, even after recovering from the acute illness [67]. There is concern that EVALI may increase the long-term risk of chronic lung diseases like COPD and pulmonary fibrosis. Ongoing research is crucial to fully elucidate the long-term health consequences of EVALI and identify risk factors for persistent respiratory complications [3].

## 13. Ongoing Research Efforts

Research efforts are ongoing to understand EVALI’s pathogenesis further, identify novel biomarkers for early detection, and develop effective prevention and treatment strategies. Studies are investigating the role of specific chemicals and additives in e-cigarette liquids in EVALI development [45]. Researchers are also exploring the impact of vaping behaviors and device characteristics on EVALI risk [39]. Advanced imaging techniques, such as MRI, can assess lung perfusion and identify subtle changes in lung structure and function in EVALI patients [82]. Clinical trials are underway to evaluate the efficacy of novel therapies, including targeted anti-inflammatory agents and lung-protective therapies [9].

## 14. Conclusions

EVALI is a serious and potentially life-threatening acute pulmonary illness that is strongly linked to the use of certain e-cigarette or vaping products. It was particularly associated with those containing vitamin E acetate, an additive commonly found in illicitly sourced THC-containing vaping liquids during the 2019–2020 outbreak. Although the overall incidence has declined significantly since that time, the potential for acute lung injury from specific constituents in vaping aerosols remains a significant public health concern. This is distinct from the better-characterized long-term risks associated with chronic nicotine e-cigarette use. Health care providers should maintain a high index of suspicion for EVALI in patients presenting with respiratory symptoms and a history of e-cigarette use. They should counsel patients on the risks associated with vaping and provide support for smoking cessation. By working together, healthcare professionals, public health officials, researchers, and policymakers can effectively address the challenges posed by EVALI and protect the respiratory health of future generations.

## Figures and Tables

**Figure 1 ijerph-22-00792-f001:**
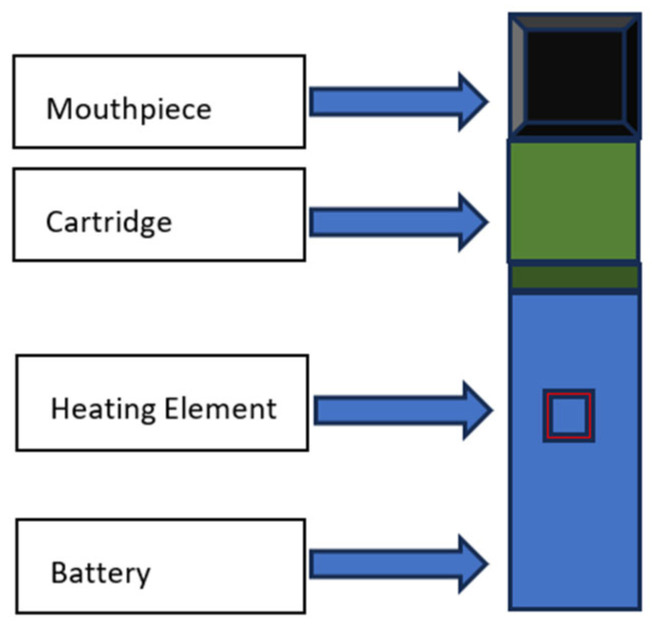
Generalized structure of a typical e-cigarette device. Components include a mouthpiece for inhalation; a cartridge or tank that holds the e-liquid; a heating element (atomizer or coil), which is powered by the battery to vaporize the e-liquid; and the battery unit. The specific design, size, and arrangement of these components can vary significantly across different types of e-cigarette devices, such as vape pens, mods, and pod systems.

**Figure 2 ijerph-22-00792-f002:**
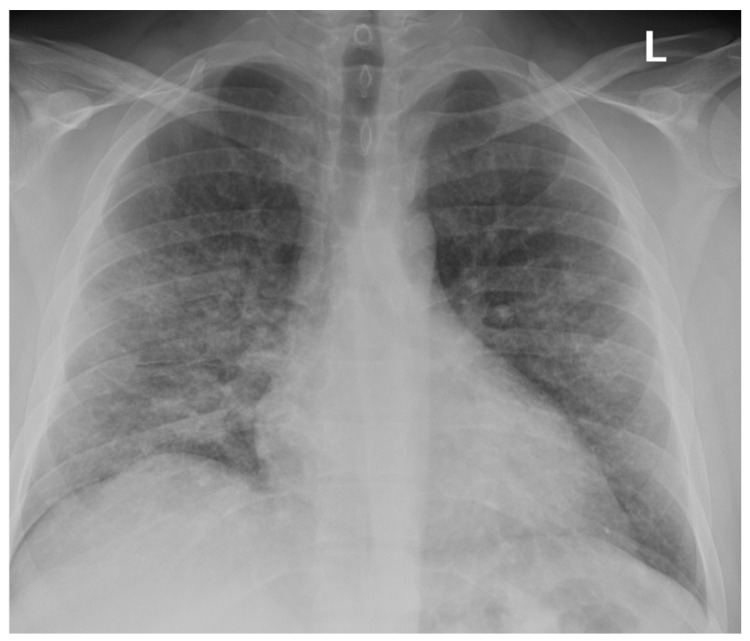
CXR showing b/l alveolar opacities.

**Figure 3 ijerph-22-00792-f003:**
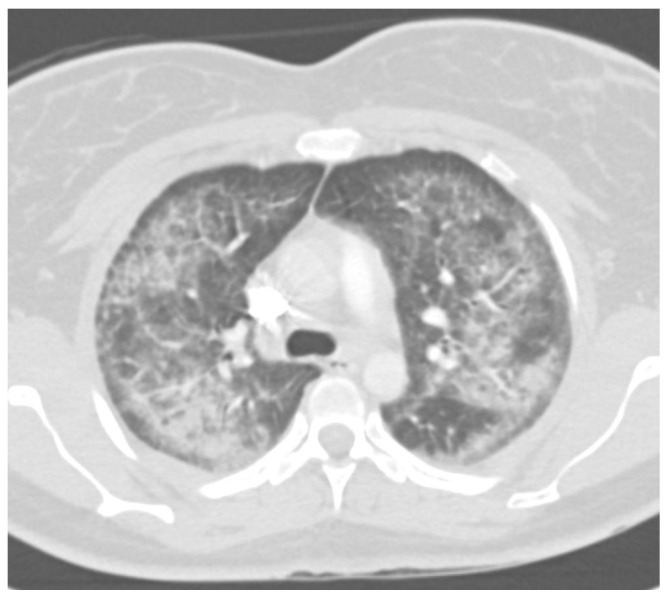
CT scans of the chest showing bilateral dense consolidated opacities with subpleural sparing on axial views.

**Figure 4 ijerph-22-00792-f004:**
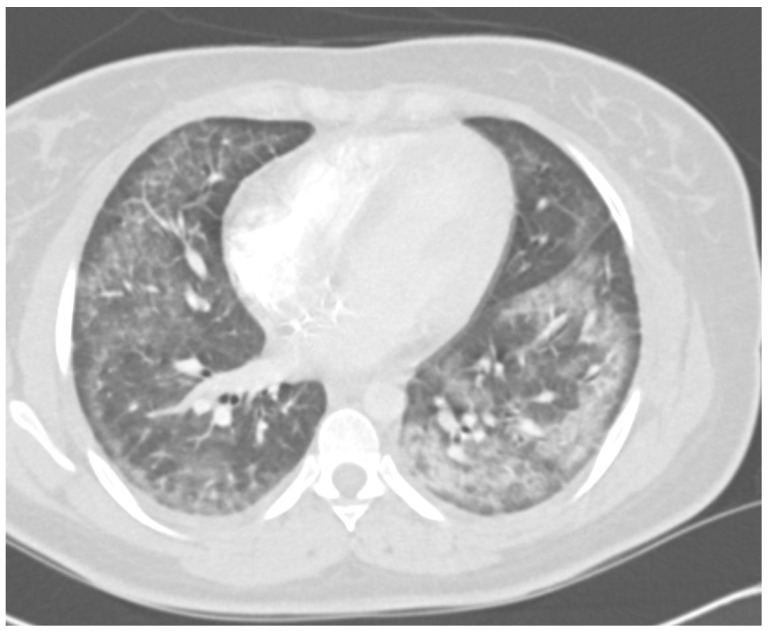
CT scans of chest showing consolidated opacities bllaterally on axial views.

**Figure 5 ijerph-22-00792-f005:**
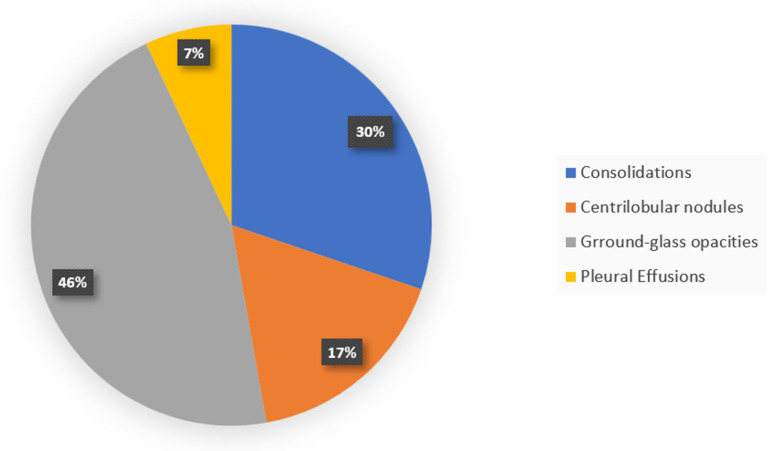
Common high-resolution computed tomography (HRCT) findings in EVALI patients [64].

**Figure 6 ijerph-22-00792-f006:**
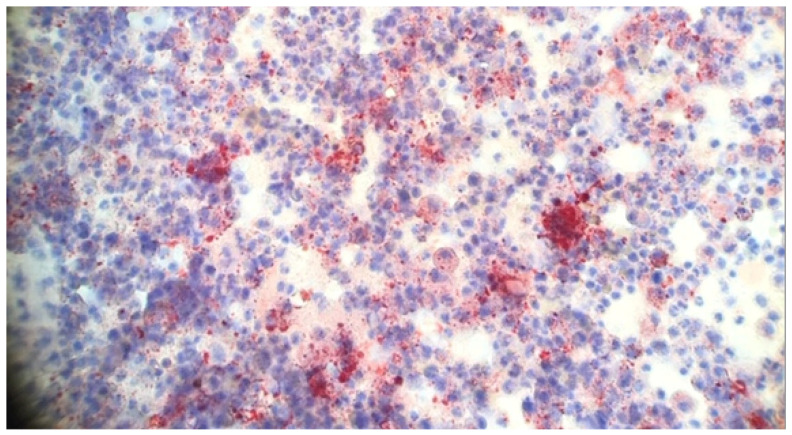
BAL cytology of a patient with EVALI showing inflammatory cells and positive oil red O staining consistent with lipid laden macrophages (100×).

**Table 1 ijerph-22-00792-t001:** Pulmonary complications associated with e-cigarettes or vaping product use.

Pulmonary Complication	Description	References
Lipoid Pneumonia	A condition caused by the accumulation of lipids (fats) in the lungs.	[6,19,20]
Hypersensitivity Pneumonia	An immune-mediated inflammatory lung disease caused by the inhalation of certain antigens.	[21,22,23]
Acute Eosinophilic Pneumonia	A rare disease characterized by the rapid accumulation of eosinophils in the lungs.	[5,24,25]
Respiratory Bronchiolitis-Associated Interstitial Lung Disease (RB-ILD)	Lung disease primarily affects the small airways (bronchioles) and the surrounding lung tissue.	[26,27,28]
Acute Fibrinous Organizing Pneumonia	A pattern of lung injury characterized by intra-alveolar fibrin deposition and organizing pneumonia.	[6,28,29]
Organizing Pneumonia	A pattern of lung inflammation and fibrosis that can occur in response to various lung injuries.	[30,31,32]
EVALI	E-cigarette or Vaping Product Use-Associated Lung Injury; a serious condition characterized by respiratory symptoms and lung damage.	[4,5,6]

**Table 3 ijerph-22-00792-t003:** Pathophysiological mechanisms.

Pathophysiological Mechanism	Key Features	Adverse Effects	References
Direct Cellular Injury and Inflammation	Exposure of lung epithelium to harmful chemicals Inflammation triggered by components like menthol, ethyl maltol, and volatile organic compounds Cellular apoptosis through ROS-mediated autophagy	General lung irritation, inflammation and increased risk of emphysema.	[48,49,50]
Bronchiolitis Obliterans	Chronic airway-centric inflammation and fibrosis Bronchiolar smooth muscle hypertrophy Peribronchiolar inflammation Intraluminal mucus accumulation Fibrotic scarring	Severe and irreversible lung damage, difficulty breathing, coughing, wheezing.	[51,52,53]
Heavy Metal Toxicity	Presence of heavy metals (chromium, nickel, lead) in e-cigarette aerosols Potential risk factors for respiratory diseases (bronchitis, asthma, COPD, lung cancer)	Increased risk of bronchitis, asthma, COPD, and lung cancer.	[16,54,55]
Lipid-Laden Alveolar Macrophages	Accumulation of lipid-laden alveolar macrophages (foam cells) in bronchoalveolar lavage fluid Dysregulation of lipid metabolism and immune homeostasis	Disrupted lung function and immune response, potential respiratory problems.	[22,45,56]
Disruption of the Alveolar-Capillary Barrier	Damage to alveolar epithelial cells and pulmonary vascular endothelial cells Fluid accumulation in the alveoli (pulmonary edema) Neutrophil infiltration Release of cytotoxic and pro-inflammatory mediators Respiratory distress	Pulmonary edema (fluid in the lungs), respiratory distress, difficulty breathing.	[25,49,57]
Vitamin E Acetate and Surfactant Dysfunction	Interference with pulmonary surfactant function Alveolar collapse and inflammation	Alveolar collapse, inflammation, impaired lung function.	[6,26,58]
Immune Dysregulation	Impairment of respiratory immune cell function Oxidative stress and inflammation Tissue damage and impaired lung function	Weakened respiratory immune system, increased susceptibility to infections, increased risk of lung damage.	[3,43,44]

**Table 4 ijerph-22-00792-t004:** CDC case definitions for EVALI (adapted from [5,79]).

Feature	Confirmed Case of EVALI	Probable Case of EVALI
E-cigarette use	Used within 90 days before symptoms started	Used within 90 days before symptoms started
Lung abnormalities	Present (seen on chest X-ray or CT scan)	Present (seen on chest X-ray or CT scan)
Infection testing	No signs of lung infection after initial tests (must include negative respiratory viral panel, influenza test if needed, and any other relevant tests based on the situation)	An infection was found but does not fully explain the lung problems, or not enough infection testing is performed, and the doctors believe the infection is not the leading cause.
Other causes	No other likely explanations for the lung problems (like heart, autoimmune, or cancer-related issues)	No other likely explanations for the lung problems (like heart, autoimmune, or cancer-related issues)

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
