# Peer review of "E-Cigarette or Vaping Product Use-Associated Lung Injury: A Comprehensive Review"

_ijerph, 2025, doi:10.3390/ijerph22050792_

Round 1

Reviewer 1 Report

Comments and Suggestions for Authors

Overall I found that this review had good content. However, the structure can use some improvement. The tables do not seem to align with the text and they are not described in test. The structure of the tables should follow logically from the text and the reader should be able to follow the tables while reading the text.  The tables are also lacking in detail and many have no citations.

I have some suggestions for improvements.

  • There are quite a few abbreviations used in this review. Please provide a list of abbreviations.
  • Please define EVALI at the first use in the first paragraph. The authors should also provide a general description of what EVALI is. Throughout the authors refer to EVALI and lung damage interchangeably. Are they same thing or is EVALI a special case of lung damage. A bit more context is needed.
  • Should EVALI not be listed in Table 1?
  • Line 77 lists some components of e-cigarettes including aldehydes, ketones and hydrocarbons. Please comment on their risks and how they related to the various illnesses provided in the tables
  • Line 81: please elaborate on the impacts to cardiovascular health
  • Lines 86-87: are these devices the same as that in Figure 1? If they are different some comment should be made
  • Section 4: Do these components lead to EVALI? The lack of clarity here may be due to a lack of a clear definition of EVALI early on.
  • Lines 124-127: Maybe there should be a separate section for cannabis. How do cannabis vapes compare to nicotine vapes?
  • Line 144: please provide the year of the citation
  • Line 144-151: this is a long sentence. Also what is the empirical evidence to support these two lines of thought?
  • Sections 6-12 are quite short. Maybe they can be combined into one section with subheadings and some more context as to how they are related to the illnesses.
  • Lines 262-278: should this not be moved to the section on long-term consequences of EVALI (lines 375 non)?
  • Please avoid the use of contractions throughout

Tables

  • Table 1 needs citations and a brief description of what the illnesses are. If possible, some reference as to their possible relationship with e-cigarettes.
  • Table 2: Please list the function of each of these items. Also these should be described in the text.
  • Table 3 needs citations. This table does not seem to align with the text. Maybe structure the table so it is easier to follow when reading the text (e.g. subheadings in the text should align with the rows in the table). This table should be paralleld in the text.  Similarly, Table 4 just seems to stand out and does not clearly align with the text. Citations are also needed for Table 4

Author Response

We thank the reviewer for the thoughtful comments and suggestions. All the reviewers comments have been answered as follows: 

Reviewer 1 comments: 

Comment#1:

There are quite a few abbreviations used in this review. Please provide a list of abbreviations:

Response: A comprehensive list of abbreviations added separately. This document added separately.

Comment #2: Please define EVALI at the first use in the first paragraph. The authors should also provide a general description of what EVALI is. Throughout the authors refer to EVALI and lung damage interchangeably. Are they same thing or is EVALI a special case of lung damage. A bit more context is needed.

Response: EVALI definition added to the introduction.

EVALI is severe manifestation of vaping-induced lung damage, not simply synonymous with all lung damage. This distinction is clarified.

Comment# 3: Should EVALI not be listed in Table 1?

Response: Yes, added to the table.

Comment#4: Line 77 lists some components of e-cigarettes including aldehydes, ketones and hydrocarbons. Please comment on their risks and how they related to the various illnesses provided in the tables

Response: While we focus on the main intentionally added components of e-cigarettes, such as nicotine, propylene glycol/vegetable glycerin, and flavorings, we acknowledged the presence and significant contribution of byproducts like aldehydes, ketones, and hydrocarbons to overall toxicity.

Comment# 5: Line 81: please elaborate on the impacts to cardiovascular health

Response: Revised the section and elaborated.

Comment # 6: Lines 86-87: are these devices the same as that in Figure 1? If they are different some comment should be made

Response: The variety of e-cigarettes share the basic components shown in Figure 1. While they differ in power and delivery we are just focusing on the fundamental mechanisms and components involved that apply across all devices. We have clarified this in the text.

Comment # 7:  Do these components lead to EVALI? The lack of clarity here may be due to a lack of a clear definition of EVALI early on.

Response: The connection between the components discussed in Section 4 and EVALI is revised and clarified.

Comment #8: Lines 124-127: Maybe there should be a separate section for cannabis. How do cannabis vapes compare to nicotine vapes?

Response: We have added few lines explaining the difference between nicotine vapes and cannabis vapes.

Comment # 9: Line 144: please provide the year of the citation

Response: The citation has been added. 

Comment # 10: Line 144-151: this is a long sentence. Also what is the empirical evidence to support these two lines of thought?

Response: The sentence has been revised, shortened and modified. 

Comment # 11: Sections 6-12 are quite short. Maybe they can be combined into one section with subheadings and some more context as to how they are related to the illnesses.

Response: Updated as sub-section of Section 5 as how the mechanisms relate to illnesses.

Comment # 12: Lines 262-278: should this not be moved to the section on long-term consequences of EVALI (lines 375 non)?Please avoid the use of contractions throughout

Response: We acknowledge the suggestion to move lines 262-278 to the long-term consequences section. However, these lines focus on radiological findings crucial for EVALI diagnosis, distinct from chronic outcomes. We maintain their current placement to preserve the radiological discussion. 

The contractions have been removed. 

Comment# 13: Table 1 needs citations and a brief description of what the illnesses are. If possible, some reference as to their possible relationship with e-cigarettes.

Response: Citations and descriptions added and this is given as a separate work document attached.

Comment # 14: Table 2: Please list the function of each of these items. Also these should be described in the text.

Response: This has been updated to show the function of each of these items and citations. 

Comment # 15: Table 3 needs citations. This table does not seem to align with the text. Maybe structure the table so it is easier to follow when reading the text (e.g. subheadings in the text should align with the rows in the table). This table should be paralleld in the text. 

Response: Revised to align with the text and citations added. A separate word document attached.

Comment # 16: 

Similarly, Table 4 just seems to stand out and does not clearly align with the text. Citations are also needed for Table 4

Response:Revised to align with the text and citations added.

Reviewer 2 Report

Comments and Suggestions for Authors

I have attached comments in document

Comments on the Quality of English Language

I have attached comments in document

Author Response

Response to reviewer # 2

We thank the reviewer for the thoughtful comments and suggestions. All the reviewers’ comments have been answered as follows: 

Reviewer 2 comments: 

Comment# 1: Line 21: The use of the term “ hazards associated with e-cigarette use” might seem biased since investigations are still underway to determine the harmful effects of e-cigarettes, particularly for those transitioning from combustible cigarettes. It would be more neutral to phrase it as “ risks which may be associated with e-cigarette use.”

Response #1: Thank you for this suggestion. We agree and have revised the phrasing to 'risks which may be associated with e-cigarette use,' as suggested.

Comment # 2: Line 34:  It is unconventional to cite a table in the introduction of a manuscript. The introduction should outline the objectives within the context of existing research. Instead, the complications could be directly listed in the text.

Response # 2: We have moved the table 1 citation from the introduction and rephased the paragraph as suggested.

Comment # 3: Line 38: Use “ two-thirds” instead of 2/3 rd for a more formal presentation

Response #3: This has been corrected as suggested.

Comment #4: The introduction lacks a mention of Vitamin E acetate, which was identified as a primary toxicant in EVALI cases. It is important to clarify that this substance was predominantly found in marijuana- vaping devices.

Response # 4: We have rephrased the introduction to reflect this and have included Vitamin E acetate

Comment # 5: Several other studies have detailed the pathology, features and management . Examples- Cherian SV, et al. E cigarette or Vaping Product Associated Lung Injury: A Review.

Response # 5: The study citated is the work of the corresponding and senior author of this manuscript and is an invited manuscript

Reviewer 3 Report

Comments and Suggestions for Authors

In the current review “E-Cigarette or Vaping, Product Use Associated Lung Injury 2 (EVALI): A Comprehensive Review ” the author aimed to investigate and compile the complexities of EVALI, including its epidemiology, pathogenesis, the structure and composition of e-cigarettes, clinical and radiological manifestations, management strategies, and public health implications. I believe this review is a systematic compilation of all necessary aspects of the topic. It would be great if the author could add a few lines about 1) treating EVALI through supplemental oxygen noninvasive ventilation, mechanical ventilation, and sometimes extracorporeal membrane oxygenation; and 2) some guidelines for the healthcare providers

Author Response

Response to reviewer # 3

We thank the reviewer for the thoughtful comments and suggestions. All the reviewers’ comments have been answered as follows: 

Reviewer 3 comments: 

Comment# 1: It would be great if the author could add a few lines about 1) treating EVALI through supplemental oxygen noninvasive ventilation, mechanical ventilation, and sometimes extracorporeal membrane oxygenation; and 2) some guidelines for the healthcare providers

Response #1: We have enhanced the clinical management section with more detail on management including specifics on oxygen titration and guidelines for healthcare providers.

Reviewer 4 Report

Comments and Suggestions for Authors

In the objective, indicate whether it is a literature review or a systematic review.
If it is a systematic review, it must follow the PRISMA criteria.

Author Response

We thank the reviewer for their comment. We confirm that this manuscript is a comprehensive literature review, not a systematic review.

Reviewer 5 Report

Comments and Suggestions for Authors

The manuscript provides a comprehensive review of EVALI, covering its epidemiology, pathogenesis, clinical features, diagnosis, management, and public health implications. The topic is timely and relevant to public health given the increased prevalence of e-cigarette use, particularly among youth.

The methodology of this review is primarily narrative rather than systematic. While the manuscript synthesizes a substantial body of literature on EVALI, there is no clearly described methodology section detailing the search strategy, inclusion/exclusion criteria, or quality assessment of included references. This makes it difficult to assess the comprehensiveness or potential selection bias in the literature reviewed.

Introduction

  • Clarify the scope of the review by explicitly stating the objectives (rather than giving examples “such as…”).

Epidemiology

  • Authors stated, “incidence of 67 EVALI peaked in mid-September 2019, followed by a rapid decline”. Are there any reported numbers on the incidence thereafter? Do cases continue or disappear?

Structure and Chemistry

  • Figure 1 needs a proper caption and higher resolution. The parts seem to be overlapping or misplaced e.g. Heating Element is shown as if it is in the battery part which is not correct.

Diagnosis and Pathology

  • Figure 5: image quality is poor.

Conclusion

  • Authors stated, “EVALI is a serious and potentially life-threatening pulmonary illness directly linked to e-cigarette and vaping product use.” The evidence presented in the article indicated that EVALI is directly linked to specific constituents rather than e-cigarette and vaping product in general.

General comments

  • Ensure all abbreviations (e.g., BAL) are defined at first use.
  • Many articles have been published on the topic in 2024 and 2025. These are not reflected in the manuscript.

Author Response

We thank the reviewer for their comprehensive feedback.

Introduction: 

Clarify the scope of the review by explicitly stating the objectives (rather than giving examples “such as…”).

Response #1 : We have clarified that this manuscript is a literature review, not a systematic review, and have added a brief statement to the introduction outlining our literature search approach, which now includes peer-reviewed articles, consensus statements, and reports published up to 2025. The Introduction has been revised to state the review's objectives.

EpidemiologyAuthors stated, “incidence of 67 EVALI peaked in mid-September 2019, followed by a rapid decline”. Are there any reported numbers on the incidence thereafter? Do cases continue or disappear?

Response #2 : The Epidemiology section has been updated with available information on EVALI incidence post the 2019-2020 outbreak. Cases still continue sporadically. 

Structure and Chemistry Figure 1 needs a proper caption and higher resolution. The parts seem to be overlapping or misplaced e.g. Heating Element is shown as if it is in the battery part which is not correct.

Response # 3: Figure 1 (e-cigarette structure) has been replaced with a higher-resolution, correctly labeled version and an improved caption.

Diagnosis and Pathology: Figure 5: image quality is poor.

Response # 4: Figure 5 (HRCT findings) has also been replaced with a higher-quality image.

Conclusion Authors stated, “EVALI is a serious and potentially life-threatening pulmonary illness directly linked to e-cigarette and vaping product use.” The evidence presented in the article indicated that EVALI is directly linked to specific constituents rather than e-cigarette and vaping product in general.

Response #5: The conclusion has been refined to more accurately reflect that EVALI is linked to specific constituents in some vaping products rather than all e-cigarette use generally.

General comments

Ensure all abbreviations (e.g., BAL) are defined at first use.Many articles have been published on the topic in 2024 and 2025. These are not reflected in the manuscript.

Response # 6: Finally, we have meticulously checked the manuscript to ensure all abbreviations are defined at their first use and have integrated findings from recent literature (up to 2025) throughout the relevant sections and updated the reference list accordingly.